# The Accumulation of Biomass Pre- and Post-Silking Associated with Gains in Yield for Both Seasons under Maize–Rice Double Cropping System

Yuling Han [1], Dong Guo [1], Fei Xia [1], Wei Ma [1], Akram Salah [2], Ming Zhan [2], Cougui Cao [2], Ming Zhao [1], Chuanyong Chen [3,*] and Baoyuan Zhou [1,*]

[1] Key Laboratory of Crop Physiology & Ecology, Ministry of Agriculture and Rural Affairs, Institute of Crop Science, Chinese Academy of Agricultural Sciences, Beijing 100081, China; hyl_0211@126.com (Y.H.); fredguod@163.com (D.G.); 13164631860@163.com (F.X.); mawei02@caas.cn (W.M.); zhaoming@caas.cn (M.Z.)

[2] MOA Key Laboratory of Crop Physiology, Ecology and Cultivation (The Middle Reaches of Yangtze River), College of Plant Science and Technology, Huazhong Agricultural University, Wuhan 430070, China; akramsalah@luawms.edu.pk (A.S.); zhanming@mail.hzau.edu.cn (M.Z.); ccgui@mail.hzau.edu.cn (C.C.)

[3] Maize Research Center, Beijing Academy of Agriculture & Forestry Sciences, Beijing 100097, China

[*] Correspondence: chuanyongchen@126.com (C.C.); zhoubaoyuan@caas.cn (B.Z.); Tel.: +86-13426014986 (C.C.); +86-15120084553 (B.Z.)

**Abstract:** Due to relatively low yield as well as low resources use efficiency with double rice (*Oryza sativa* L.) cropping systems (RR), exploring new cropping systems to increase yield and resources use efficiency simultaneously has become a large challenge of the middle reaches of the Yangtze River (MRYR). Our previous study demonstrated that the maize (*Zea mays* L.)–rice cropping system (MR) exhibited higher superiority of yield and resource use efficiency compared with the conventional double-rice cropping system. However, the reason for the yield increases in both maize and rice and the physiological processes involved in those two crops under MR are poorly understood. A 3-year field experiment was conducted at two sites (Wuxue and Jingmen) from 2016 to 2018 to examine the differences in dry matter (DM) accumulation, soil properties, and resources use efficiency between the MR and RR cropping systems. Compared with RR, the annual yield of MR was 18.2–26.3% and 15.4–31.5% higher across three years at Wuxue and Jingmen, respectively. The average yield of maize in MR was 36.5% and 21.9% higher than that of early rice in RR at Wuxue and Jingmen, respectively. The yield increase for maize was mainly attributed to the 29.7% (Wuxue) and 28.5% (Jingmen) increases in post-silking DM accumulation due to the higher plant growth rate promoted by the higher net assimilation rate and radiation use efficiency. For the late rice in MR, the average yield was 10.9% and 14.5% higher than that of late rice in RR at Wuxue and Jingmen, respectively, which was promoted by the 7.8–23.3% increase in pre-anthesis DM accumulation due to improved soil properties. Compared with RR, the MR cropping system exhibited increased soil pH, total organic carbon, and mineral nitrogen, and decreased the bulk density in the late rice season. As a result of greater yield in both seasons under MR, the annual accumulated temperature and radiation use efficiency, partial factor productivity from applied nitrogen, and water use efficiency of MR were 17.7–26.4%, 22.2–25.5%, 5.5–7.8%, and 33.6–48.7% higher than those of RR, respectively. We conclude that the higher yield in the MR than in the RR cropping system was mainly attributed to the accumulation of post-silking biomass due to maximized use of radiation in the maize season, and the accumulation of pre-anthesis biomass due to improved soil nutrients in the late rice season.

**Keywords:** maize–rice cropping system; grain yield; dry matter accumulation; resource use efficiency; soil properties

## 1. Introduction

The middle reaches of the Yangtze River (MRYR) are the main grain-producing area in China and an important multiple-crop farming region rich in radiation and temperature resources, where the double-rice (*Oryza sativa* L.) (RR) cropping system was the major cropping system [1]. However, successive years of rice–rice continuous cropping in paddies have resulted in many production and environmental problems. Long-term paddy cultivation increases greenhouse gas (especially $CH_4$) emissions [2–4], and increases the soil compaction, which limits nutrient absorption and root development [5]. In addition, frequently occurring seasonal drought caused by global climate change increases the risk of yield loss for rice [6], and conventional puddle-transplanted rice involves high energy and labor [7]. Therefore, it is necessary to explore new cropping systems that achieve high yields, high resource use efficiency, and minimal environmental impact in the MRYR.

Farmers are searching for alternate sustainable cropping system options that can enhance crop yield while reducing production costs [8]. Because of the high resource use efficiency of maize (*Zea mays* L.), a $C_4$ crop, introducing maize into rice-based systems provides a new maize–rice cropping system (MR), which can simultaneously increase crop yields and resource use efficiency in the MRYR. Compared with rice, maize transports assimilation products more efficiently and distributes the material and energy provided by photosynthesis, which is beneficial for annual yields in the MR cropping system. In addition, the drivers for replacing early rice with spring maize also include (a) the increasing demand for maize in the poultry sector and the tightening world export–import market [9–12]; (b) the changes in soil structure and increased soil quality resulting from the introduction of maize in rotation with rice [13]; (c) and the reduced use of pesticides and herbicides with strong adaptability following the earlier sowing and harvesting of maize than early rice with high mechanization, which increased farmers income [14,15]. At present, the MR planted area exceeds $3 \times 10^6$ ha, and MR has become the predominant choice for the diversification of existing rice-based crop rotations in Asia [16].

Previous studies found that the MR cropping system had high grain yield but also high nutrient demand [10]. However, a paddy–upland rotation can help improve soil properties and increase soil nutrients [17], increase the amount of soil bacteria and microorganisms, and improve the distribution of soil bacteria and fungi compared with the paddy field cropping system [18]. The increased soil nutrient content from dry season crop growth and harvest in the rice-based double cropping system could provide a good foundation for the growth and yield formation of late rice [19]. Therefore, the implementation of appropriate management practices, such as straw return [20] and balanced chemical fertilizer rates [21], could further result in high crop productivity in MR cropping systems with less fertilizer application due to the increased total and available soil nutrition. Moreover, the MR cropping system could significantly reduce carbon emissions and has a lower carbon footprint [22], lower water, labor, and energy requirements, and provides a higher net income compared with the conventional RR system [23–25]. In our previous study, we found that the yields of the maize and rice seasons in the MR cropping system were significantly higher than those of the corresponding seasons in the RR cropping system, which resulted in a significantly higher annual yield and resource use efficiency for MR than for RR. However, the reasons for the increased yield in both crops and the physiological processes between the maize and rice in MR are poorly understood. The present study aimed to (1) evaluate the differences in dry matter (DM) accumulation, grain yield, and radiation use efficiency between spring maize and early rice; (2) examine the effect of paddy–upland rotation on soil microorganisms, soil nutrients, and N accumulation in late rice; and (3) compare the accumulated temperature and radiation use efficiency, partial factor productivity from applied nitrogen, and water use efficiency between MR and RR cropping system. The results of these investigations will help to clarify the mechanisms of yield increases for both maize and rice under the MR cropping system and will provide guidance for further improving crop production, soil properties, and environmental conditions in the MRYR and similar agricultural regions worldwide.

## 2. Materials and Methods

### 2.1. Experimental Sites

The field experiments were conducted from 2016 to 2018 at Wuxue (30°01′ N, 115°74′ E) and Jingmen (30°52′ N, 112°50′ E), Hubei Province, China. Wuxue and Jingmen counties are typical areas with a humid mid-subtropical monsoonal climate in the MRYR. They are intensive agricultural areas, where the double-rice cropping system has been the dominant planting system in the past. The two experimental sites had different levels of soil fertility. The main soil properties (0–20 cm depth) in Wuxue were as follows: the pH was 7.0 (extracted by $H_2O$; soil: water = 1:2.5), the bulk density (BD) was 1.27 g cm$^{-3}$, the organic carbon (TOC) concentration was 14.25 g kg$^{-1}$, the total nitrogen (TN) concentration was 1.56 g kg$^{-1}$, the total phosphorus (P) concentration was 0.52 g kg$^{-1}$, the Olsen P concentration was 9.56 mg kg$^{-1}$, the exchangeable potassium (K) concentration (extracted by $CH_3COONH_4$) was 93.72 mg kg$^{-1}$, and the mineral nitrogen ($N_{min}$) concentration was 21.08 mg kg$^{-1}$. The soil properties (0–20 cm depth) in Jingmen were as follows: the pH was 7.0, the BD was 1.27 g cm$^{-3}$, the organic C concentration was 14.07 g kg$^{-1}$, the total N concentration was 1.49 g kg$^{-1}$, the total P concentration was 0.52 g kg$^{-1}$, the Olsen P concentration was 13.49 mg kg$^{-1}$, the exchangeable K concentration was 200 mg kg$^{-1}$, and the $N_{min}$ concentration was 8.98 mg kg$^{-1}$.

### 2.2. Experimental Design and Cropping Management

A randomized complete block design was employed, with two treatments and three replications at two sites. The treatments were the maize–rice and double-rice cropping systems. Each plot was 8 m × 20 m in size, with an area of 160 m$^2$. The main plots were surrounded by ridges 25 cm in height. Strong black plastic film was driven into the soil along the inner edge of the field ridge to a depth of 30 cm and used to cover the entire ridge to prevent water movement among plots. The two seasons' crop sowing, heading, and harvest dates are shown in Table 1.

**Table 1.** Two cropping systems with different planting density, sowing, flowering, and harvest date under the first and second seasons at Wuxue and Jingmen.

| Year | Site | Treatment | First Season | | | Second Season | | |
|------|------|-----------|--------------|--------------|--------------|--------------|--------------|--------------|
| | | | Sowing Date | Heading Date | Harvest Date | Sowing Date | Heading Date | Harvest Date |
| 2016 | Wuxue | MR | 9 March | 24 May | 14 July | 24 June | 12 September | 24 October |
| | | RR | 22 March | 7 June | 17 July | 24 June | 12 September | 24 October |
| | Jingmen | MR | 25 March | 1 June | 15 July | 20 June | 14 September | 29 October |
| | | RR | 20 March | 5 June | 18 July | 20 June | 15 September | 1 November. |
| 2017 | Wuxue | MR | 12 March | 26 May | 17 July | 24 June | 12 September | 25 October |
| | | RR | 25 March | 11 June | 19 July | 24 June | 12 September | 25 October |
| | Jingmen | MR | 23 March | 3 June | 13 July | 22 June | 14 September | 2 November |
| | | RR | 20 March | 7 June | 20 July | 23 June | 20 September | 3 November |
| 2018 | Wuxue | MR | 8 March | 24 May | 13 July | 21 June | 9 September | 22 October |
| | | RR | 27 March | 13 June | 20 July | 24 June | 14 September | 24 October |
| | Jingmen | MR | 24 March | 1 June | 14 July | 21 June | 15 September | 1 November |
| | | RR | 19 March | 5 June | 17 July | 20 June | 16 September | 2 November |

MR: maize–rice cropping system; RR: double-rice cropping system.

In the MR cropping system, the maize cultivar Zhengdan958 and the late rice cultivar Huanghuazhan, which are widely grown in the MRYR, were used at both experimental sites in three years. The experimental field was plowed and prepared before seeding spring maize. In the spring maize season, nitrogen (240 kg N ha$^{-1}$ as urea), phosphorus (135 kg $P_2O_5$ ha$^{-1}$ as calcium superphosphate), and potassium (180 kg $K_2O$ ha$^{-1}$ as KCl) were applied at Wuxue. Meanwhile, nitrogen (240 kg N ha$^{-1}$ as urea), phosphorus (90 kg $P_2O_5$ ha$^{-1}$ as calcium superphosphate), and potassium (135 kg $K_2O$ ha$^{-1}$ as KCl) were applied in the spring maize season at Jingmen. At both sites, 30% N (nitrogen), 50% K (phosphorus), and 100% P (potassium) were applied as a basal fertilizer before maize sowing. Of the remaining N, 30% was applied at the 6-leaf stage, and 40% at the 12-leaf stage, and the remaining 50% K was applied at the 12-leaf stage in the spring maize season. Only natural rainfall was used in the spring maize season, and without artificial irrigation. Bed furrows with beds 1.0 m in width and furrows 0.2 m in width were

formed to alleviate the impacts of waterlogging. Maize seeds were sown in early March manually in three rows on the bed with two seeds per hole. The wide and narrow row spacing was 60 cm and 27 cm, respectively. The plants were thinned at the three-leaf stage to a stand density of $6 \times 10^4$ plants ha$^{-1}$ each year. The late rice was sown during late June in the nursery, and the rice seedlings were manually transplanted after the maize harvest. After the maize was harvested, the plots were soaked with water for 3 days and subsequently plowed and puddled using the rotary tiller. Then, the late rice seedlings were transplanted into the maize–rice treatment plots. The transplanting density for late rice was $33.35 \times 10^4$ hills ha$^{-1}$ with 25 cm and 12 cm row and hill spacing each year, respectively. In the annual late rice season, 240 kg ha$^{-1}$ N, 105 kg ha$^{-1}$ P, and 160 kg ha$^{-1}$ K were applied at Wuxue, and 240 kg ha$^{-1}$ N, 75 kg ha$^{-1}$ P, and 135 kg ha$^{-1}$ K were applied at Jingmen, respectively. At both sites, 100% P, 40% N, and 50% K were applied as basal fertilizers before late rice transplanting. A total of 20% of the N was applied during the tillering stage. At the booting stage, 40% of the N and 50% of the K were applied. After the late rice was transplanted, 2 cm water layer was maintained in the field, and field water was cut off from rice milking to maturity stage. In addition to rainfall, the rice field was irrigated several times. The amount of irrigation water ranged from 30 to 500 mm based on soil moisture. Irrigation water was applied via a 15 cm plastic hose. A flow meter recorded the amount of irrigation water used.

In the RR cropping system, the early rice cultivar E'zao8 and the late rice cultivar Huanghuazhan, which are widely grown in the MRYR, were used at both experimental sites in three years. Early rice was sown during late March in the nursery, and the rice seedlings were transplanted in late April. The transplanting density for early rice was $33.35 \times 10^4$ hills ha$^{-1}$ with 25 cm and 12 cm row and hill spacing each year, respectively. In the annual early rice season, 180 kg ha$^{-1}$ N, 105 kg ha$^{-1}$ P, and 180 kg ha$^{-1}$ K were applied at Wuxue and 180 kg ha$^{-1}$ N, 75 kg ha$^{-1}$ P, and 135 kg ha$^{-1}$ K were applied at Jingmen, respectively. At both sites, 100% P, 40% N, and 50% K were applied as basal fertilizers before late rice transplanting. A total of 20% of the N was applied during the tillering stage. At the booting stage, 40% of the N and 50% of the K were applied. All the agronomic management practices were the same as the above-mentioned description for late rice in the MR cropping system. After the early rice seedlings were transplanted, the agronomic management practices applied were identical to those used for the late rice. All agronomic management practices applied were the same as those described for late rice in the MR cropping system.

### 2.3. Measurements Methods

#### 2.3.1. Weather Data

Daily weather data for the experimental site (daily mean temperature, daily maximum and minimum temperatures, precipitation, and sunshine hours) during the crop growing seasons from 2016 to 2018 were obtained from the Chinese Meteorological Administration (2018).

Solar radiation was calculated as follows [26]:

$$\text{Solar radiation } Q = Q0 \left(a + b \frac{S}{S_0}\right) \tag{1}$$

where Q is the total solar radiation, $Q_0$ is the astronomical radiation, S is the actual number of sunshine hours, $S_0$ is the possible number of sunshine hours, $S/S_0$ is the proportion of actual sunshine hours to the possible number of sunshine hours, and a and b are the correction coefficients.

Growing degree days (GDD, °C d$^{-1}$) was calculated as follows [27]:

$$\text{GDD} = \sum_0^n \left[ \frac{T_{max} + T_{min}}{2} \right] - T_{base} \tag{2}$$

where GDD is the number of active growing degree days; n is the number of consecutive periods during the threshold temperature period; and $T_{max}$ (maximum temperature), $T_{min}$ (minimum temperature), and $T_{base}$ are the maximum and minimum daily temperatures and the base temperature of 10 °C, respectively.

### 2.3.2. Soil Properties

In 2017 and 2018, after late rice harvest at two sites, five soil samples were taken in each plot at 20 cm depth and then mixed. A 2 mm mesh filter was used as a sieve for the fresh soil samples, which were then split into two parts. One part was stored at 4 °C for the mineralized nitrogen ($N_{min}$) and pH assessments, and the other was air-dried to assess the TOC and TN contents. The BD (g cm$^{-3}$) of the 0–20 cm soil layer was measured via a coring method [28]. The soil pH was determined with a pH meter at a 1:2.5 soil: water ratio (*w/v*). A CHNOS elemental analyzer (Vario MAX, Elementar, Germany) was used for the TOC and TN measurements after the samples were passed through a 150 µm mesh screen.

### 2.3.3. Net Assimilation Rate

The net assimilation rate (g m$^{-2}$ d$^{-1}$) for maize and rice was calculated as follows [29]:

$$\text{Net assimilation rate} = \frac{\ln \text{LAI}_2 - \ln \text{LAI}_1}{\text{LAI}_2 - \text{LAI}_1} \times \frac{W_2 - W_1}{t_2 - t_1} \tag{3}$$

where $\text{LAI}_1$ and $\text{LAI}_2$ are the calculated leaf area index, $W_1$ and $W_2$ are the calculated DM in the crop flowering and maturity stage, and $t_1$ and $t_2$ are the date of crop flowering and maturity stage, respectively.

### 2.3.4. Dry Matter Accumulation

At silking/anthesis and maturity, 20 maize plants and 6 rice hills in each plot were randomly sampled and oven-dried at 85 °C to a constant weight to obtain the aboveground DM mass. The post-silking/anthesis DM is calculated by the following:

$$\text{DM at post-silking/anthesis (Mg ha}^{-1}) = \text{DM at maturity} - \text{DM at pre-silking/anthesis} \tag{4}$$

Plant growth rate (PGR) is defined as

$$\text{PGR} \left( \text{kg ha}^{-1} \text{ d}^{-1} \right) = \frac{\text{DM}}{\text{D}} \tag{5}$$

where D means crop growing days to produce the DM.
DM producing energy is defined as

$$\text{DM producing energy (MJ m}^{-2}) = \text{DM} \times \text{GCV} \tag{6}$$

GCV (J g$^{-1}$) is the gross caloric value, and means the energy released by the complete combustion of per gram of DM. In our experiment, the GCV of both maize and rice is $1.779 \times 10^4$ J g$^{-1}$ [6].

### 2.3.5. Grain Yield and Yield Components

At harvesting, rice plants were sampled diagonally across three sample areas of 3 m$^2$ in each plot in which the grain yield and yield components were determined. The number of panicles for each plant within 1.0 m$^2$ was counted. In addition, the number of filled spikelets per panicle was recorded as the mean spikelet number of 20 ears from each replication, while 1000 grain weight was calculated from the average of five random samples of 500 grains. The maize grain yield and yield components were determined from 50 adjacent maize plants in the middle rows of each plot. The kernel number per ear was recorded as the mean kernel number of 20 ears from each replication, while the 1000-kernel

weight of maize was measured from the average of five random samples of 500 kernels. The final yields for maize and rice were adjusted to the standard moisture content (14%).

2.3.6. Resources Use Efficiency

The production efficiency of radiation ($R_a$) and accumulated temperature (AT) were calculated as follows:

$$\text{Production efficiency of } R_a \left( g\ MJ^{-1} \right) = \frac{GY}{R_a} \tag{7}$$

$$\text{Production efficiency of AT} \left( kg\ ha^{-1}\ {}^\circ C^{-1} \right) = \frac{GY}{AT} \tag{8}$$

where GY is the grain yield, $R_a$ is the radiation per unit area, and AT is the accumulated degree days (where the base temperature is 10 °C).

The $R_a$ utilization efficiency (RUE) was calculated as follows [30]:

$$RUE = \frac{H \times W}{\sum Q} \times 100\% \tag{9}$$

where RUE is the radiation utilization efficiency, H is the conversion coefficient of DM to heat energy (for maize and rice, the value of H is 0.01807 MJ $g^{-1}$), W is the actual maize or rice DM obtained in the experimental period, and $\sum Q$ (MJ $m^{-2}$) is the total radiation during the maize and rice growing season.

The irrigation water use efficiency (WUE, kg $m^{-3}$) for the grain yield (kg $ha^{-1}$) was calculated as follows [31]:

$$WUE = \frac{GY}{I} \tag{10}$$

where GY is the grain yield (kg $ha^{-1}$) and I is the water consumption (mm).

The partial factor productivity from applied N ($PFP_N$) is the ratio of the grain yield to the applied N rate, which is calculated as follows [32]:

$$PEP_N = \frac{GY}{F} \tag{11}$$

where F is the N rate (kg $ha^{-1}$) applied in the treatments.

2.3.7. Economic Benefits

The net economic return was calculated as follows [31]:

$$\text{Net economic return} = \text{Gross revenue from grain yield–Costs} \tag{12}$$

where costs includes the costs of fertilizers, seeds, labor, machinery, herbicides and pesticides, and mulching film. These costs were based on the average local market prices from 2016 to 2018.

*2.4. Statistical Analysis*

Analyses of variance were performed using the Statistix 8.0 statistical package. The results are given as the means of three replicates, and the differences between treatments were determined by comparing their means with the least significant difference (LSD) test at the 0.05 probability level.

**3. Results**

*3.1. Weather Conditions*

The monthly total precipitation and average temperature during the maize and rice growth seasons was different at Wuxue and Jingmen from 2016 to 2018 (Figure 1). At Wuxue, the effective accumulative temperature ($\geq$10 °C) was 2790.6, 2791.7, and 2867.5 °C in 2016,

2017, and 2018, respectively. The precipitation was 1530.5, 1208.6, and 977.5 mm in 2016, 2017, and 2018, respectively. The radiation value in the maize season was 4094.7, 4361.5, and 4614.0 MJ m$^{-2}$ in 2016, 2017, and 2018, respectively. At Jingmen, the accumulative temperature ($\geq$10 °C) was 2544.6, 2632.1, and 2644.0 °C in 2016, 2017, and 2018, respectively. The precipitation was 1396.9, 1189.8, and 888.1 mm in 2016, 2017, and 2018, respectively. The radiation values in the maize season were 4107.4, 4361.5, and 4612.0 MJ m$^{-2}$ in 2016, 2017, and 2018, respectively. During three experimental years, the monthly radiation was similar at Wuxue and Jingmen. Total rainfall and mean monthly temperature were higher at Wuxue than at Jingmen in three experimental years. The total precipitation in the first season was 1025.0, 762.8, and 516.8 mm at Wuxue and 856.4, 571.7, and 467.8 mm at Jingmen in 2016, 2017, and 2018, respectively. Irrespective of the experimental sites, monthly mean temperature in March, June, and September were 1.5, 1.5, and 1.2 °C higher on average in 2018 than in 2017, respectively, while the rainfall at Wuxue and Jingmen was 98.6 and 134.8 mm less in 2018 than in 2017, respectively.

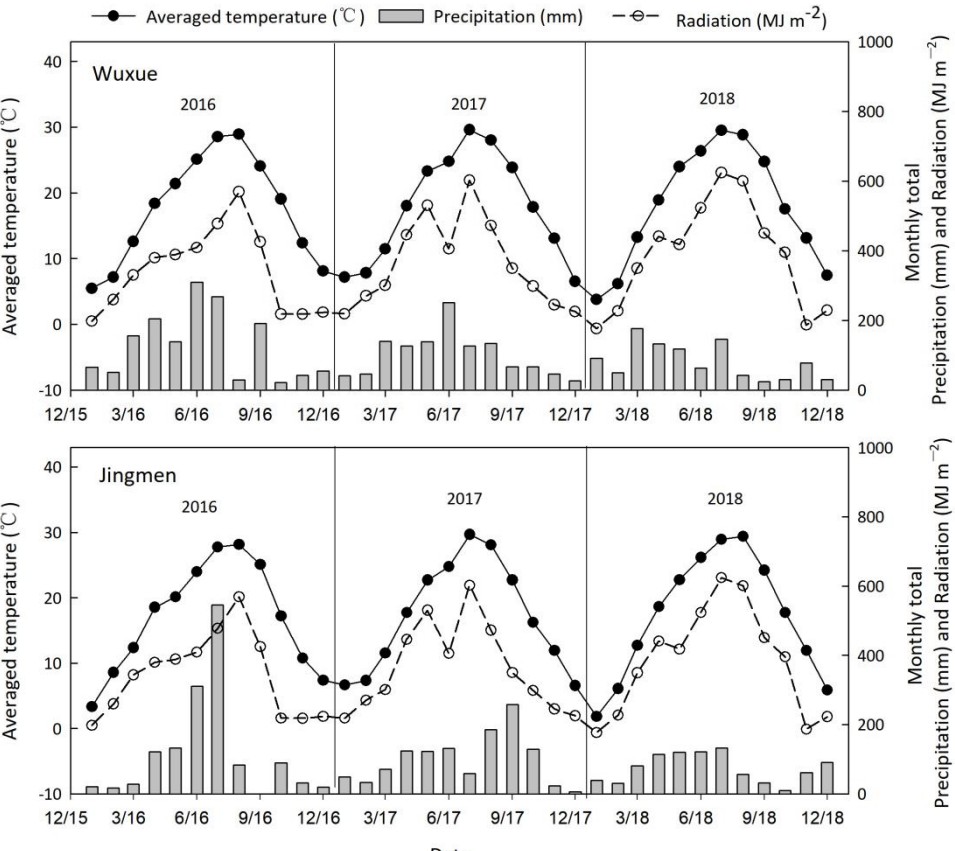

**Figure 1.** Monthly averaged temperature and total precipitation and radiation at Wuxue and Jingmen from 2016 to 2018.

### 3.2. Grain Yield and Yield Components

The seasonal and annual grain yields were both significantly affected by the experimental sites, years, and cropping systems (Table 2). The annual grain yield under MR was significantly higher than that under RR at Wuxue and Jingmen from 2016 to 2018 (Figure 2). At Wuxue, the annual yields under the MR cropping system were 18.9, 18.1, and 19.7 Mg ha$^{-1}$, which were 18.2%, 25.4%, and 26.3% higher than those of RR in 2016, 2017, and 2018, respectively. In Jingmen, the MR cropping system significantly increased the annual grain yield by 15.4%, 31.5%, and 16.3% compared with RR in 2016, 2017, and 2018, respectively. The maize and rice yields under MR both showed significant increasing trends in comparison with the early rice and late rice yields under RR at both sites from 2016 to 2018. At Wuxue, the maize yields under MR were 9.5, 10.4, and 10.7 Mg ha$^{-1}$, which were

33.5%, 34.8%, and 41.3% higher than those of early rice under RR in 2016, 2017, and 2018, respectively. In addition, the yields of late rice under MR were 9.4, 7.7, and 9.1 Mg ha$^{-1}$, which were 5.8%, 14.7%, and 12.3% higher than those of late rice under RR in 2016, 2017, and 2018, respectively. At Jingmen, the grain yields of maize under MR were 18.5%, 50.8%, and 16.3% higher than those of early rice under RR in 2016, 2017, and 2018, respectively. In addition, the grain yields of late rice under MR were 12.5%, 14.6%, and 16.3% higher than those of late rice under RR in 2016, 2017, and 2018, respectively.

**Table 2.** Analysis of variance for the crop yields in the first and second seasons and annually from 2016 to 2018 at Wuxue and Jingmen, China.

| Source of Variation | First Season | Second Season | Annual |
|---|---|---|---|
| Site (S) | ** | ** | ** |
| Year (Y) | ** | ** | ** |
| Treatment (T) | ** | ** | ** |
| S × Y | ** | ** | ** |
| S × T | ** | NS | ** |
| Y × T | ** | NS | ** |
| S × Y × T | ** | NS | * |

Levels of significance indicated: ** $p \leq 0.01$, * $p \leq 0.05$, and NS = not significant.

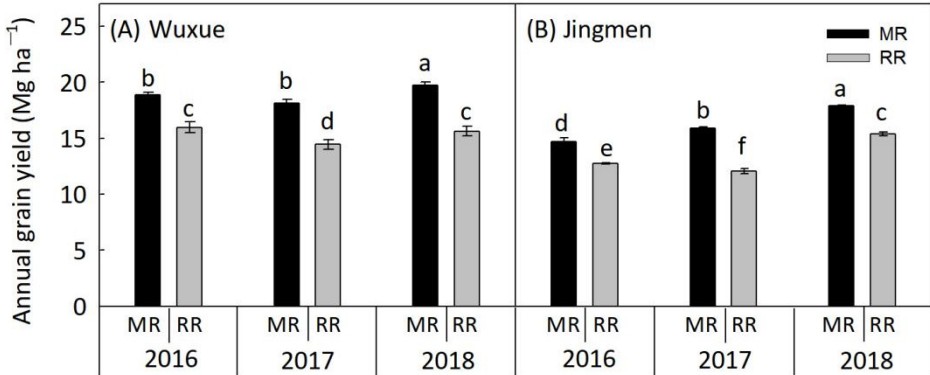

**Figure 2. Annual** grain yield for MR and RR cropping system at Wuxue (**A**) and Jingmen (**B**) from 2016 to 2018. Different letters indicate significant differences between values within the same column among 3 years according to the least significant difference (LSD) test ($p < 0.05$). MR: maize–rice cropping system; RR: double-rice cropping system.

Yield components of spring maize and rice in 2017 and 2018 at Wuxue are shown in Table 3. The 1000-kernel weight and kernel number of maize in 2018 were significantly higher than in 2017. Rice in 2018 also showed significantly higher 1000-grain weight than that in 2017. The late rice in both MR and RR cropping systems in 2018 showed significantly higher panicles compared with those in 2017.

### 3.3. DM Accumulation, DM Producing Energy, and Net Assimilation Rate

The DM accumulation and producing energy in the first and second seasons and the annual aboveground biomass differed significantly between experimental sites, years, and cropping systems. The DM accumulation and producing energy in the first and second seasons and the annual aboveground biomass under the MR cropping system were significantly higher than those under the RR cropping system at both sites (Table 4). Compared with RR, MR increased the annual DM accumulation and producing energy by 20.2%, 21.5%, and 18.3% at Wuxue and by 17.4%, 24.2%, and 20.3% at Jingmen in 2016, 2017, and 2018, respectively. At Wuxue, the DM accumulation and producing energy of maize under MR was 29.7% higher than that of early rice under RR, and the DM accumulation and producing energy of late rice under MR was 10.8% higher than that of late rice under RR. At Jingmen, the DM accumulation and producing energy of maize under MR was significantly higher by 31.2% than that of early rice under RR, and the DM accumulation and producing energy of late rice under MR was 10.9% higher than that of late rice under RR.

**Table 3.** Yield and yield components of maize and rice in 2017 and 2018 at Wuxue.

| Year | Treatment | Maize | | | | Rice | | | |
|------|-----------|-------|-------|-------|-------|------|------|------|------|
| | | Ear Number (Ha$^{-1}$) | 1000-Kernel Weight (G) | Kernel Number (Ear$^{-1}$) | Grain Yield (Mg Ha$^{-1}$) | Spikelet Panicle$^{-1}$ | 1000-Grain Weight (G) | Panicles (M$^{-2}$) | Grain Yield (Mg Ha$^{-1}$) |
| 2017 | MR-M | 84,317.94 a | 287.15 b | 410.91 b | 10.40 a | - | - | - | - |
| | MR-R | - | - | - | - | 133.77 ab | 21.73 b | 259.19 bc | 7.71 bc |
| | RR-1 | - | - | - | - | 130.53 ab | 21.79 b | 257.07 bc | 7.71 bc |
| | RR-2 | - | - | - | - | 129.80 b | 21.21 b | 238.71 c | 6.72 d |
| 2018 | MR-M | 83,664.31 a | 308.32 a | 429.03 a | 10.68 a | - | - | - | - |
| | MR-R | - | - | - | - | 136.44 a | 23.28 a | 300.66 a | 9.06 a |
| | RR-1 | - | - | - | - | 131.37 ab | 22.74 a | 257.64 bc | 7.56 c |
| | RR-2 | - | - | - | - | 132.39 ab | 22.90 a | 281.67 ab | 8.07 b |

Data are means ± standard error (*n* = 3). Different letters within the same column indicate significant differences among the rice in different cropping system in 2017 and 2018 at Wuxue according to the least significant difference (LSD) test (*p* < 0.05). MR-M: spring maize in the maize–rice cropping system; MR-R: late rice in the maize–rice cropping system; RR-1: early rice in the double-rice cropping system; RR-2: late rice in the double-rice cropping system.

**Table 4.** Dry matter accumulation and producing energy under the MR and RR cropping systems from 2016 to 2018 at Wuxue and Jingmen.

| Year | Treatment | Dry Matter Accumulation (Mg Ha$^{-1}$) | | | | | | Dry Matter Producing Energy (MJ M$^{-2}$) | | | | | |
|------|-----------|---------|---------|---------|---------|---------|---------|---------|---------|---------|---------|---------|---------|
| | | Wuxue | | | Jingmen | | | Wuxue | | | Jingmen | | |
| | | 1st Season | 2nd Season | Annual | 1st Season | 2nd Season | Annual | 1st Season | 2nd Season | Annual | 1st Season | 2nd Season | Annual |
| 2016 | MR | 19.40 ± 0.39 b | 18.81 ± 0.39 a | 38.21 ± 0.52 b | 17.10 ± 0.40 b | 15.50 ± 0.26 b | 32.60 ± 0.49 b | 34.51 ± 0.70 b | 33.47 ± 0.70 a | 67.98 ± 0.92 b | 30.42 ± 0.72 b | 27.59 ± 0.46 b | 32.60 ± 0.49 b |
| | RR | 14.97 ± 0.43 d | 16.82 ± 0.29 c | 31.78 ± 0.19 d | 13.32 ± 0.16 d | 14.46 ± 0.27 cd | 27.78 ± 0.23 d | 26.63 ± 0.77 d | 29.92 ± 0.52 c | 56.55 ± 0.33 d | 23.70 ± 0.29 d | 25.73 ± 0.48 cd | 27.78 ± 0.23 d |
| 2017 | MR | 21.78 ± 0.23 a | 17.97 ± 0.51 ab | 39.75 ± 0.74 ab | 18.08 ± 0.28 ab | 14.94 ± 0.16 bc | 33.02 ± 0.29 b | 38.75 ± 0.40 a | 31.98 ± 0.91 ab | 70.73 ± 1.32 ab | 32.16 ± 0.49 ab | 26.59 ± 0.29 bc | 33.02 ± 0.29 b |
| | RR | 16.39 ± 0.51 c | 16.31 ± 0.22 c | 32.70 ± 0.66 cd | 12.64 ± 0.52 d | 13.95 ± 0.25 d | 26.59 ± 0.77 d | 29.17 ± 0.90 c | 29.02 ± 0.39 c | 58.19 ± 1.18 cd | 22.49 ± 0.93 d | 24.82 ± 0.44 d | 26.59 ± 0.77 d |
| 2018 | MR | 21.05 ± 0.64 a | 18.84 ± 0.51 a | 39.89 ± 1.15 a | 18.65 ± 0.77 a | 17.18 ± 0.27 a | 35.83 ± 1.04 a | 37.46 ± 1.14 a | 33.52 ± 0.91 a | 70.98 ± 2.05 a | 33.19 ± 1.37 a | 30.56 ± 0.48 a | 35.83 ± 1.04 a |
| | RR | 16.64 ± 0.15 c | 17.08 ± 0.12 bc | 33.72 ± 0.26 c | 15.28 ± 0.31 c | 14.51 ± 0.59 cd | 29.79 ± 0.28 c | 29.60 ± 0.27 c | 30.39 ± 0.22 bc | 60.00 ± 0.47 c | 27.18 ± 0.56 c | 25.82 ± 1.05 cd | 29.79 ± 0.28 c |
| Source of variation | | | | | | | | | | | | | |
| Year (Y) | | ** | * | * | ** | ** | ** | ** | * | * | ** | ** | ** |
| Cropping system (C) | | ** | ** | ** | ** | ** | ** | ** | ** | ** | ** | ** | ** |
| Y × C | | NS | NS | NS | * | ** | NS | NS | NS | NS | * | ** | NS |

Data are means ± standard error (*n* = 3). Levels of significance indicated: ** *p* ≤ 0.01, * *p* ≤ 0.05, and NS = not significant. Different letters within the same column indicate significant differences among treatments from 2016 to 2018 at the same crop stage and site according to the least significant difference (LSD) test (*p* < 0.05). MR: maize–rice cropping system; RR: double-rice cropping system.

The seasonal pre- and post-silking/anthesis DM and plant growth rate (PGR) at Wuxue and Jingmen are shown in Tables 5 and 6. The post-silking DM and PGR of maize were significantly affected by cropping system and by the interaction of year and cropping system, while the post-anthesis DM and PGR of rice were affected by the cropping system at Wuxue and Jingmen, respectively. At Wuxue (Table 5), there were no significant differences in the pre-silking/anthesis DM and PGR of maize in MR and those of early rice in the RR cropping system, on average, over the 3 years. During the post-silking period, the DM and PGR of maize increased by 53.7% and 23.0%, respectively, compared with those of early rice in RR, on average, over the 3 years. In each year, the DM and PGR of maize also increased by 39.3% and 11.5% in 2016, 72.9% and 38.4% in 2017, and 48.9% and 19.1% in 2018 compared with those of early rice in RR, respectively. Significant differences in the DM and PGR of late rice during pre-anthesis for late rice in the MR and RR cropping systems were found across 3 years. At Jingmen (Table 6), the post-silking PGR of maize under MR was increased by 43.9% compared with early rice in RR averaged over 3 years. During pre-anthesis, the DM and PGR of late rice in MR increased by 13.6% and 14.9% compared with those of late rice in the RR cropping system, respectively, averaged over 3 years. During post-anthesis, the PGR of late rice in MR also significantly increased by 8.9% in 2016 and 31.3% in 2018 compared with that in late rice in RR.

**Table 5.** Dry matter, growth duration, and plant growth rate (PGR) during pre- and post-silking under maize in MR and pre- and post-anthesis under rice under MR and RR treatments in Wuxue from 2016 to 2018.

| | | First Season | | | | | | Second Season | | | | | |
|---|---|---|---|---|---|---|---|---|---|---|---|---|---|
| | | Pre-Silking/Anthesis | | | Post-Silking/Anthesis | | | Pre-Anthesis | | | Post-Anthesis | | |
| Year | Treatment | Dry Matter (Mg Ha$^{-1}$) | Duration (D$^{-1}$) | PGR (Kg Ha$^{-1}$ D$^{-1}$) | Dry Matter (Mg Ha$^{-1}$) | Duration (D$^{-1}$) | PGR (Kg Ha$^{-1}$ D$^{-1}$) | Dry Matter (Mg Ha$^{-1}$) | Duration (D$^{-1}$) | PGR (Kg Ha$^{-1}$ D$^{-1}$) | Dry Matter (Mg Ha$^{-1}$) | Duration (D$^{-1}$) | PGR (Kg Ha$^{-1}$ D$^{-1}$) |
| 2016 | MR | 8.21 ± 0.13 a | 63 | 130.27 ± 1.99 a | 11.19 ± 0.27 b | 50 | 223.79 ± 5.32 b | 8.86 ± 0.34 b | 73 | 121.33 ± 4.62 b | 9.95 ± 0.26 a | 42 | 236.97 ± 6.29 a |
| | RR | 6.93 ± 0.26 b | 66 | 105.07 ± 3.97 b | 8.03 ± 0.17 c | 40 | 200.77 ± 4.30 c | 8.22 ± 0.15 cd | 73 | 112.56 ± 2.11 cd | 8.60 ± 0.30bc | 42 | 204.73 ± 7.22 bc |
| 2017 | MR | 8.64 ± 0.32 a | 63 | 137.08 ± 5.13 a | 13.14 ± 0.44 a | 50 | 262.88 ± 8.85 a | 8.73 ± 0.30 bc | 73 | 119.64 ± 4.07 bc | 9.24 ± 0.30 b | 42 | 219.93 ± 7.18 b |
| | RR | 8.79 ± 0.47 a | 66 | 133.22 ± 7.19 a | 7.60 ± 0.22 c | 40 | 190.00 ± 5.48 c | 7.98 ± 0.25 d | 73 | 109.30 ± 3.39 d | 8.33 ± 0.13 c | 42 | 198.37 ± 2.98 c |
| 2018 | MR | 8.58 ± 0.34 a | 63 | 136.18 ± 5.45 a | 12.48 ± 0.58 a | 50 | 249.51 ± 11.52 a | 10.17 ± 0.12 a | 73 | 139.34 ± 1.65 a | 8.66 ± 0.47 bc | 42 | 206.30 ± 11.21 bc |
| | RR | 8.26 ± 0.57 a | 66 | 125.16 ± 8.56 a | 8.38 ± 0.58 c | 40 | 209.44 ± 14.41 bc | 9.08 ± 0.13 b | 73 | 124.44 ± 1.72 b | 8.00 ± 0.06 c | 42 | 190.42 ± 1.47 c |
| Year (Y) | | ** | - | ** | NS | - | NS | ** | - | ** | ** | - | ** |
| Cropping system (C) | | NS | - | ** | ** | - | ** | ** | - | ** | ** | - | ** |
| Y × C | | NS | - | NS | ** | - | ** | NS | - | NS | NS | - | NS |

Data are means ± standard error ($n$ = 3). Levels of significance indicated: ** $p \leq 0.01$, and NS = not significant. Different letters within the same column indicate significant differences among treatments from 2016 to 2018 at the same crop stage and site according to the least significant difference (LSD) test ($p < 0.05$). MR: maize–rice cropping system; RR: double-rice cropping system.

**Table 6.** Dry matter, growth duration, and plant growth rate (PGR) during pre- and post-silking under maize in MR and pre- and post-anthesis under rice under MR and RR treatments in Jingmen from 2016 to 2018.

| | | First Season | | | | | | Second Season | | | | | |
|---|---|---|---|---|---|---|---|---|---|---|---|---|---|
| | | Pre-Silking/Anthesis | | | Post-Silking/Anthesis | | | Pre-Anthesis | | | Post-Anthesis | | |
| Year | Treatment | Dry Matter (Mg Ha$^{-1}$) | Duration (D$^{-1}$) | PGR (Kg Ha$^{-1}$ D$^{-1}$) | Dry Matter (Mg Ha$^{-1}$) | Duration (D$^{-1}$) | PGR (Kg Ha$^{-1}$ D$^{-1}$) | Dry Matter (Mg Ha$^{-1}$) | Duration (D$^{-1}$) | PGR (Kg Ha$^{-1}$ D$^{-1}$) | Dry Matter (Mg Ha$^{-1}$) | Duration (D$^{-1}$) | PGR (Kg Ha$^{-1}$ D$^{-1}$) |
| 2016 | MR | 8.36 ± 0.13 b | 62 | 134.79 ± 2.08 bc | 8.74 ± 0.28 b | 45 | 194.21 ± 6.13 b | 7.72 ± 0.15 bc | 79 | 97.78 ± 1.91 bc | 7.78 ± 0.18 b | 45 | 172.88 ± 3.92 b |
| | RR | 7.59 ± 0.09 c | 66 | 115.01 ± 1.42 d | 5.73 ± 0.07 d | 43 | 133.18 ± 1.63 d | 7.12 ± 0.21 de | 80 | 89.44 ± 2.58 de | 7.34 ± 0.06 bc | 46 | 159.54 ± 1.33 c |
| 2017 | MR | 9.12 ± 0.14 a | 62 | 147.17 ± 2.23 a | 8.95 ± 0.14 b | 45 | 198.94 ± 3.06 b | 8.51 ± 0.07 a | 79 | 107.71 ± 0.88 a | 6.43 ± 0.10 d | 45 | 142.95 ± 2.32 d |
| | RR | 6.70 ± 0.28 d | 66 | 101.48 ± 4.20 e | 5.94 ± 0.25 cd | 43 | 138.13 ± 5.72 cd | 6.90 ± 0.33 e | 80 | 86.24 ± 4.07 e | 7.05 ± 0.49 cd | 46 | 153.32 ± 10.75 cd |
| 2018 | MR | 8.67 ± 0.02 ab | 62 | 139.84 ± 0.39 ab | 9.98 ± 0.76 a | 45 | 221.84 ± 16.84 a | 8.25 ± 0.24 ab | 79 | 104.45 ± 3.04 ab | 8.93 ± 0.14 a | 45 | 198.36 ± 3.17 a |
| | RR | 8.55 ± 0.32 b | 66 | 129.57 ± 4.80 c | 6.73 ± 0.52 c | 43 | 156.42 ± 12.08 c | 7.56 ± 0.29 cd | 80 | 94.48 ± 3.66 cd | 6.95 ± 0.37 cd | 46 | 151.11 ± 8.15 cd |
| Year (Y) | | ** | - | ** | ** | - | ** | NS | - | NS | ** | - | ** |
| Cropping system (C) | | ** | - | ** | ** | - | NS | ** | - | * | ** | - | ** |
| Y × C | | ** | - | ** | NS | - | NS | * | - | * | ** | - | ** |

Data are means ± standard error ($n$ = 3). Levels of significance indicated: ** $p \leq 0.01$, * $p \leq 0.05$, and NS = not significant. Different letters within the same column indicate significant differences among treatments from 2016 to 2018 at the same crop stage and site according to the least significant difference (LSD) test ($p < 0.05$). MR: maize–rice cropping system; RR: double-rice cropping system.

The net assimilation rates in the first season, second season, and annually in MR were significantly higher than those in RR at Wuxue in 2017 and 2018 (Figure 3). Maize had the highest net assimilation rate among the crops in the MR and RR cropping systems in 2017 and 2018. The net assimilation rate of maize was 207.6% and 197.6% higher than that of early rice under RR in 2017 and 2018, respectively. No significant difference in the net assimilation rate of late rice between the MR and RR cropping systems at Wuxue was found in 2017 or 2018. In addition, the annual net assimilation rate of MR was 128.9% and 76.6% higher than that of RR in 2017 and 2018, respectively.

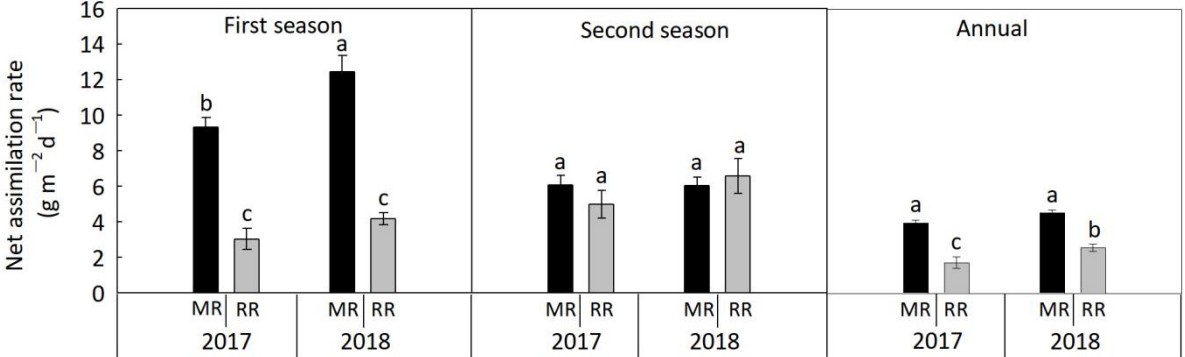

**Figure 3.** Net assimilation rate for crops under MR and RR cropping systems at Wuxue from 2017 to 2018. Different letters indicate significant differences between values within the same column in the same season according to the least significant difference (LSD) test ($p < 0.05$). MR: maize–rice cropping system; RR: double-rice cropping system.

### 3.4. Soil Bulk Density, TOC, and $N_{min}$ Content

The soil BD, pH, TOC, and Nmin were all affected by cropping systems on both soil types at Wuxue and by year at Jingmen (Table 7). At Wuxue, the concentration of soil TOC and Nmin under the MR cropping system were 5.5% and 7.9%, and 4.9% and 10.9% higher than those under RR in 2017 and 2018, respectively. In addition, soil BD under the MR cropping system was 3.9% and 5.4% lower than that under RR in 2017 and 2018, respectively. The soil pH under the MR cropping system was 9.6% higher than that under RR in 2018. At Jingmen, the concentration of soil TOC under the MR cropping system was 5.6% higher than that under RR in 2018. In addition, the soil Nmin under MR was 13.5% and 17.2% higher than that under RR in 2017 and 2018, respectively. The soil BD of the MR cropping system was 5.2% and 4.5% lower than that of RR in 2017 and 2018, respectively.

**Table 7.** Soil properties after the first crop harvest at both experimental sites in 2017 and 2018.

| Year | Treatment | Wuxue | | | | Jingmen | | | |
|---|---|---|---|---|---|---|---|---|---|
| | | BD | pH | TOC | $N_{min}$ | BD | pH | TOC | $N_{min}$ |
| 2017 | MR | 1.23 ± 0.01 b | 6.53 ± 0.02 a | 15.05 ± 0.13 a | 23.12 ± 0.45 a | 1.27 ± 0.00 b | 6.91 ± 0.12 a | 14.83 ± 0.11 a | 11.20 ± 0.53 a |
| | RR | 1.28 ± 0.02 a | 6.32 ± 0.03 b | 14.27 ± 0.10 b | 21.49 ± 0.45 b | 1.34 ± 0.00 a | 6.54 ± 0.06 b | 14.18 ± 0.19 a | 9.87 ± 0.24 b |
| 2018 | MR | 1.22 ± 0.01 b | 6.86 ± 0.07 a | 15.33 ± 0.08 a | 24.86 ± 0.40 a | 1.28 ± 0.01 b | 6.90 ± 0.07 a | 15.02 ± 0.22 a | 12.27 ± 0.62 a |
| | RR | 1.29 ± 0.00 a | 6.26 ± 0.04 b | 14.62 ± 0.07 b | 22.42 ± 0.72 b | 1.34 ± 0.01 a | 6.64 ± 0.06 b | 14.22 ± 0.12 b | 10.47 ± 0.24 b |
| Year (Y) | | NS | ** | ** | ** | NS | NS | NS | NS |
| Cropping system (C) | | ** | ** | ** | ** | ** | ** | ** | ** |
| Y × C | | NS | ** | NS | NS | NS | NS | NS | NS |

Data are means ± standard error ($n = 9$). Levels of significance indicated: ** $p \leq 0.01$, and NS = not significant. Different letters within the same column indicate significant differences between treatments in the same year and site according to the least significant difference (LSD) test ($p < 0.05$). BD: bulk density; TOC: total organic carbon; $N_{min}$: mineral nitrogen.

### 3.5. Resources Use Efficiency

The production efficiency of radiation ($R_a$) and accumulated temperature (AT) in the first and second seasons and annually were significantly affected by years and cropping systems at both sites (Table 8). Maize in MR had a higher production efficiency of $R_a$ than

early rice in RR at both sites, by 25.6% at Wuxue and 33.6% at Jingmen, on average, over 3 years. Late rice in MR also had higher production efficiency of $R_a$ than early rice in RR (except in 2016 at Wuxue) at both sites, by 11.0% at Wuxue and by 19.4% at Jingmen, on average over 3 years. The MR cropping system had a higher annual production efficiency of $R_a$ than the RR cropping system at both sites, by 17.7% at Wuxue and 26.4% at Jingmen, on average over three years.

**Table 8.** Production efficiency of radiation and accumulated temperature in the MR and RR cropping systems from 2016 to 2018 at Wuxue and Jingmen.

| Year | Treatment | Wuxue | | | | | | Jingmen | | | | | |
|---|---|---|---|---|---|---|---|---|---|---|---|---|---|
| | | Production Efficiency of $R_a$ (g MJ$^{-1}$) | | | Production Efficiency of AT (kg ha$^{-1}$ °C) | | | Production Efficiency of $R_a$ (g MJ$^{-1}$) | | | Production Efficiency of AT (kg ha$^{-1}$ °C) | | |
| | | 1st Season | 2nd Season | Annual | 1st Season | 2nd Season | Annual | 1st Season | 2nd Season | Annual | 1st Season | 2nd Season | Annual |
| 2016 | MR | 0.57 ± 0.01 a | 0.54 ± 0.02 a | 0.55 ± 0.01 a | 6.53 ± 0.07 b | 4.84 ± 0.15 a | 5.56 ± 0.06 a | 0.50 ± 0.02 b | 0.40 ± 0.00 b | 0.45 ± 0.01 a | 5.89 ± 0.23 b | 3.80 ± 0.04 c | 4.62 ± 0.10 c |
| | RR | 0.47 ± 0.01 c | 0.51 ± 0.02 a | 0.49 ± 0.01 b | 4.99 ± 0.07 cd | 4.57 ± 0.20 ab | 4.75 ± 0.14 b | 0.39 ± 0.01 d | 0.36 ± 0.00 d | 0.37 ± 0.00 b | 4.55 ± 0.08 e | 3.38 ± 0.04 e | 3.86 ± 0.02 e |
| 2017 | MR | 0.54 ± 0.01 b | 0.45 ± 0.00 b | 0.50 ± 0.01 b | 6.95 ± 0.18 a | 4.07 ± 0.02 c | 5.34 ± 0.08 a | 0.44 ± 0.01 c | 0.44 ± 0.00 a | 0.44 ± 0.00 a | 5.43 ± 0.06 c | 4.37 ± 0.01 a | 4.81 ± 0.03 b |
| | RR | 0.43 ± 0.01 d | 0.39 ± 0.01 cd | 0.41 ± 0.01 c | 5.22 ± 0.16 c | 3.55 ± 0.10 d | 4.28 ± 0.09 c | 0.31 ± 0.02 e | 0.33 ± 0.00 e | 0.32 ± 0.01 c | 3.80 ± 0.19 f | 3.31 ± 0.03 e | 3.52 ± 0.07 f |
| 2018 | MR | 0.56 ± 0.01 ab | 0.42 ± 0.01 bc | 0.48 ± 0.01 b | 6.73 ± 0.11 ab | 4.50 ± 0.06 b | 5.48 ± 0.08 a | 0.53 ± 0.00 a | 0.39 ± 0.00 c | 0.45 ± 0.00 a | 6.38 ± 0.02 a | 4.29 ± 0.02 b | 5.14 ± 0.02 a |
| | RR | 0.43 ± 0.02 d | 0.37 ± 0.01 d | 0.40 ± 0.01 c | 4.85 ± 0.17 d | 4.00 ± 0.07 c | 4.37 ± 0.12 c | 0.40 ± 0.01 d | 0.34 ± 0.00 e | 0.37 ± 0.00 b | 4.94 ± 0.08 d | 3.69 ± 0.03 d | 4.23 ± 0.05 d |
| Source of variation | | | | | | | | | | | | | |
| Year (Y) | | ** | ** | ** | * | ** | ** | ** | ** | ** | ** | ** | ** |
| Cropping system (C) | | ** | ** | ** | ** | ** | ** | ** | ** | ** | ** | ** | ** |
| Y × C | | NS | NS | NS | NS | NS | NS | NS | ** | ** | NS | ** | ** |

Data are means ± standard error ($n = 3$). Levels of significance indicated: ** $p \leq 0.01$, * $p \leq 0.05$, and NS = not significant. Different letters within the same column indicate significant differences among treatments from 2016 to 2018 at the same crop stage and site according to the least significant difference (LSD) test ($p < 0.05$). MR: maize–rice cropping system, RR: double-rice cropping system.

The MR cropping system had a higher production efficiency of AT than the RR cropping system at both sites over the 3 years (Table 8). Maize in MR had a higher production efficiency of AT than early rice in RR, by 34.2% and 33.2% at Wuxue and Jingmen, respectively, on average over 3 years. Late rice in MR also had a higher production efficiency of AT than late rice in RR (except in 2016 at Wuxue), by 49.0% and 19.4% at Wuxue and Jingmen, respectively, on average over 3 years. In addition, the annual production efficiency of AT in MR increased by 22.2% and 25.5% compared with that in RR, respectively, on average over 3 years.

The $PFP_N$ in the second season and the annual $PFP_N$ were affected by the year and cropping system at Wuxue, while the $PFP_N$ of the first and second seasons and the annual $PFP_N$ were all affected by the year and cropping system at Jingmen (Table 9). No significant difference in $PFP_N$ of maize was found between the MR and RR cropping systems at Wuxue, while the $PFP_N$ of maize under MR was 9.2% higher than that of early rice under RR. Late rice under MR had higher $PFP_N$ values than late rice under RR, by 10.6% and 20.1% at Wuxue and Jingmen, respectively, on average over 3 years. Therefore, the annual $PFP_N$ for MR was 7.8% and 5.5% higher than that for RR at Wuxue and Jingmen, respectively, on average over 3 years.

The WUE of the first and second seasons and the annual WUE were all affected by year and cropping system at both sites. The MR cropping system had a higher WUE in the first season, second season, and annually compared with that in the RR cropping system at both Wuxue and Jingmen across the three experimental years (Table 9). Maize in MR had a higher WUE, by 63.8% and 98.1%, than early rice in RR at Wuxue and Jingmen, respectively, averaged over 3 years. The WUE of the late rice under MR was 10.6% and 20.3% higher than that under RR averaged over 3 years at Wuxue and Jingmen, respectively. The annual

WUE in MR was 33.6% and 48.7% higher than that in RR, on average over 3 years at Wuxue and Jingmen, respectively.

**Table 9.** N partial productivity from applied N (PFP$_N$) and water use efficiency (WUE) in the MR and RR cropping systems from 2016 to 2018 at Wuxue and Jingmen.

| Year | Treatment | Wuxue | | | | | | Jingmen | | | | | |
|---|---|---|---|---|---|---|---|---|---|---|---|---|---|
| | | PFP$_N$ (kg kg$^{-1}$) | | | WUE (kg ha$^{-2}$ mm$^{-1}$) | | | PFP$_N$ (kg kg$^{-1}$) | | | WUE (kg ha$^{-2}$ mm$^{-1}$) | | |
| | | 1st Season | 2nd Season | Annual | 1st Season | 2nd Season | Annual | 1st Season | 2nd Season | Annual | 1st Season | 2nd Season | Annual |
| 2016 | MR | 39.65 ± 0.44 c | 38.99 ± 1.20 a | 39.32 ± 0.41 ab | 15.16 ± 0.17 b | 11.01 ± 0.34 a | 12.77 ± 0.13 b | 30.63 ± 1.21 d | 30.67 ± 0.29 d | 30.65 ± 0.68 c | 14.07 ± 0.56 c | 7.79 ± 0.07 d | 10.02 ± 0.22 c |
| | RR | 39.59 ± 0.54 c | 36.83 ± 1.61 a | 38.02 ± 1.15 bc | 8.38 ± 0.12 e | 10.40 ± 0.45 a | 9.39 ± 0.28 e | 34.46 ± 0.64 c | 27.26 ± 0.34 e | 30.35 ± 0.14 c | 5.70 ± 0.11 f | 6.92 ± 0.09 f | 6.27 ± 0.03 e |
| 2017 | MR | 43.32 ± 1.10 ab | 32.11 ± 0.19 b | 37.72 ± 0.54 bc | 13.63 ± 0.35 c | 9.07 ± 0.05 b | 11.23 ± 0.16 c | 30.76 ± 0.36 d | 35.44 ± 0.12 b | 33.10 ± 0.24 b | 15.87 ± 0.18 b | 10.01 ± 0.03 b | 12.08 ± 0.09 b |
| | RR | 42.86 ± 1.34 ab | 28.00 ± 0.80 c | 34.37 ± 0.72 d | 11.02 ± 0.35 d | 7.91 ± 0.23 c | 9.31 ± 0.20 e | 31.35 ± 1.54 d | 26.83 ± 0.28 e | 28.77 ± 0.53 d | 8.06 ± 0.40 e | 7.58 ± 0.08 e | 7.80 ± 0.14 d |
| 2018 | MR | 44.51 ± 0.75 a | 37.75 ± 0.47 a | 41.13 ± 0.61 a | 20.67 ± 0.35 a | 10.66 ± 0.13 a | 14.44 ± 0.21 a | 37.80 ± 0.11 b | 36.67 ± 0.16 a | 37.23 ± 0.12 a | 19.39 ± 0.05 a | 10.35 ± 0.04 a | 13.56 ± 0.04 a |
| | RR | 42.00 ± 1.51 bc | 33.61 ± 0.60 b | 37.20 ± 0.99 c | 10.80 ± 0.39 d | 9.49 ± 0.17 b | 10.08 ± 0.27 d | 43.33 ± 0.70 a | 31.51 ± 0.29 c | 36.58 ± 0.44 a | 11.14 ± 0.18 d | 8.90 ± 0.08 c | 9.91 ± 0.12 c |
| Source of variation | | | | | | | | | | | | | |
| Year (Y) | | ** | ** | ** | ** | ** | ** | ** | ** | ** | ** | ** | ** |
| Cropping system (C) | | NS | ** | ** | ** | ** | ** | ** | ** | ** | ** | ** | ** |
| Y × C | | NS | NS | NS | ** | NS | ** | * | ** | ** | NS | ** | * |

Data are means ± standard error ($n = 3$). Levels of significance indicated: ** $p \leq 0.01$, * $p \leq 0.05$, and NS = not significant. Different letters indicate significant differences among treatments from 2016 to 2018 within the same column at the same crop stage and site according to the least significant difference (LSD) test ($p < 0.05$). MR: maize–rice cropping system; RR: double-rice cropping system.

*3.6. Economic Benefits*

The MR cropping system had lower economic inputs and a higher net income than the RR cropping system at both Wuxue and Jingmen averaged over the three experimental years (Table 10). At Wuxue, maize under the MR cropping system produced a higher net income, by 306.5%, than early rice under the RR cropping system. In addition, the annual net income from MR was 100.7% higher than that from RR. At Jingmen, maize under the MR cropping system produced a higher net income, by 226.3%, than early rice under the RR cropping system. In addition, the annual net income under MR was 106.6% higher than that under RR.

**Table 10.** Economic input and net income for the MR and RR cropping systems at Wuxue and Jingmen over three years.

| Site | Treatment | Input (US$ ha$^{-1}$ year$^{-1}$) | | | | | | Total Input (USD ha$^{-1}$ year$^{-1}$) | Output (USD ha$^{-1}$ year$^{-1}$) | Net Income (USD ha$^{-1}$ year$^{-1}$) |
|---|---|---|---|---|---|---|---|---|---|---|
| | | Labor | Machine | Seeds | Fertilizer | Pesticides | Mulching Film | | | |
| Wuxue | MR | 985.71 | 1028.57 | 182.14 | 821.29 | 289.29 | 107.14 | 3414.14 | 6563.99 ± 235.09 a | 3149.85 ± 215.82 a |
| | RR | 1200.00 | 1157.14 | 171.43 | 752.00 | 321.43 | 0.00 | 3602.00 | 5171.16 ± 271.84 b | 1569.16 ± 249.86 b |
| Jingmen | MR | 628.57 | 642.86 | 172.99 | 703.71 | 289.29 | 107.14 | 2544.56 | 5665.19 ± 387.00 a | 3120.62 ± 387.00 a |
| | RR | 914.29 | 914.29 | 205.71 | 646.86 | 321.43 | 0.00 | 3002.57 | 4512.82 ± 422.02 b | 1510.25 ± 422.02 b |

USD 1 = CNY 7.0. Data are means ± standard error ($n = 3$). Different lowercase letters among values within the same column indicate significant differences in maize and rice seasons averaged over three experimental years according to the least significant difference (LSD) test ($p < 0.05$).

**4. Discussion**

In the MRYR, many successive years of rice–rice continuous cropping systems in paddies have resulted in several production and environmental problems, such as soil compaction, greenhouse gas emissions, and the adverse weather conditions encountered in the first season. These problems have seriously limited further increases in grain yield and resource use efficiency in this area. Recently, MR cropping systems, in which maize takes the place of early rice in the RR cropping system, have developed rapidly because of their

higher yield and benefit and improved resource utilization efficiency compared with those in the RR cropping system [1,10]. In our 3-year experiment, at both Wuxue and Jingmen, the yields of maize, late rice, and annual yield for MR increased significantly compared with the RR cropping system. These results were consistent with the previous studies showing that MR had significantly higher grain yield than the RR cropping system [1,6]. Thus, the MR system has been recommended as an alternative to double-rice cropping [33]; however, the reason for the increased yields of both maize and rice seasons and the physiological process involved under MR need to be further examined.

Crop yield is positively correlated with DM accumulation [34,35]. We found that the increases in maize and rice yield in MR were mainly attributed to the increased DM accumulation at both sites in the three years (Tables 5 and 6). In the present study, the total biomass of maize in MR was higher than that of early rice in RR due to the greater DM accumulation post-silking in maize. Since GCV (gross caloric value) is an important indicator for evaluating the solar energy accumulation and chemical energy conversion efficiency of plants, it can eliminate the influence of crop types. For this reason, we compared the differences in DM producing energy of MR and RR cropping systems. In our study, the DM accumulation and producing energy in the two cropping systems showed the same trend, that MR showed significantly higher DM producing energy compared with RR. Crop DM accumulation mainly occurs through photosynthesis, and grain yields largely depend on the utilization of radiation [36]. A study showed that the maximum RUE of $C_3$ plants is 4.6% and that of $C_4$ plants is 6% [37], mainly because $C_4$ plants have higher photosynthetic efficiency, light saturation point, and assimilation capacity than $C_3$ plants [38]. In our study, the $C_3$ crop rice was replaced by the $C_4$ crop maize, and the production efficiency of radiation and accumulated temperature for maize in MR was significantly higher than that in RR. In addition, the plant growth rate post-silking of maize reached 190.5–245.4 kg ha$^{-1}$ °C, which was significantly higher than that of early rice. Zhou et al. [39] found that the yield of maize kernels was mainly derived from photosynthetic products formed from the silking to maturity stages. The net photosynthetic efficiency of plants refers to the amount of assimilates accumulated in a unit of leaf area per unit time after subtracting plant respiration consumption [37]. In our study, the net photosynthetic efficiency of maize was significantly higher than that of early rice, which resulted in a 76.6–128.9% increase for the annual net photosynthetic efficiency of MR than that of RR (Figure 3).

At both Wuxue and Jingmen, the total biomass of the late rice in MR was higher than that of the late rice in RR, which was attributed to the greater DM accumulation pre-anthesis in rice (Tables 5 and 6). Compared with the RR cropping system, changing the paddy planting to upland planting for the first season in the MR cropping system could improve the soil properties for the next planting season. The soil bulk density after harvest of the first-season crop in MR decreased significantly compared with that in RR, which promoted the renewal of soil organic matter and increased soil nutrients for the second season in MR. The TOC and $N_{min}$ in the 0–20 cm soil layer under the late rice in MR were significantly higher than those in RR. In the different cropping systems, C inputs contain seeds, fertilizer, crop roots, litter, and rhizodeposition, and MR had a higher C input into the soil than the RR cropping system [22]. Researchers reported that drying and wetting regimes increased soil TOC and TN and increased soil $N_{min}$ content [40,41], which is similar to our result. In our study, compared with the RR system, the MR system significantly increased the TOC and $N_{min}$ after 3 years of experiments in both Wuxue and Jingmen (Table 7). An increased level of TOC is known to regulate the chemical and physical biological processes that collectively determine soil quality [8]. In our experiment, after nearly four months of rice irrigation in the MR system, the decomposition and consumption of soil organic matter were restricted in the flooded state, which promoted the accumulation of soil fertility in our experiment [42–44]. Zhang et al. [45] reported that soil fertility factors, such as TOC and $N_{min}$, are significantly related to the soil community structure in paddy and dry rotation systems. The paddy and upland system of MR will cause significant changes to the

quantity of microbial communities in soil, increase soil microbial activity, and promote the circulation and transformation of nutrients such as nitrogen, phosphorus, and potassium in soil [46,47]. In our study, the yield of late rice in MR increased by 10.9% and 14.5% compared with that of the late rice in RR averaged over three experimental years at Wuxue and Jingmen, respectively (Figure 2). This finding was consistent with the results showing that the upland planting in the first season could significantly (by 10.8%) increase the yield of late rice in the following season reported by Zhang et al. [48]. Singh et al. [8] reported that the growth and harvest of dry-season maize increases the soil nutrient content for the next season of late rice in MR cropping systems, thus laying a good fertility foundation for the development of late rice and increasing its pre-anthesis DM accumulation and yield formation.

Based on the results of the present study, the MR cropping system produced significantly higher yields of first season, second season, and annual than the RR cropping system in both Wuxue and Jingmen. Higher DM accumulation post-silking and radiation use efficiency in the maize season contributed to the higher crop yield of the first season in MR compared to that in the RR cropping system. The paddy–dry rotation system (MR) improved soil quality and increased the late rice DM accumulation pre-anthesis, resulting in a higher yield of late rice than that in RR. As a result of the increased yield in the MR cropping system, the production efficiency of $R_a$ and AT, $PEP_N$, and WUE of MR significantly increased compared with RR at the two sites. The results were confirmed by other studies where MR had not only significantly higher annual yield than RR, but the annual radiation and temperature production efficiency, water use efficiency, and solar radiation use efficiency under MR were 14.7%, 20.4%, 12.1%, and 19.1% higher than those under RR, respectively [6]. In addition, because the MR cropping system required fewer inputs and obtained higher income than that of RR cropping system, the annual net income for MR averaged over 3 years was 100.7% and 106.6% higher than that of RR at Wuxue and Jingmen, respectively. Despite these observations, some aspects of the MR cropping system, such as the environmental sustainability of MR, need to be further examined.

## 5. Conclusions

Our study demonstrated that the higher post-silking DM accumulation due to the higher plant growth rate and radiation use efficiency increased maize yield in MR compared with early rice in RR, while the higher pre-anthesis DM accumulation, promoted by the higher soil TOC and $N_{min}$ contents and abundances of two cellulose-decomposing fungi in the 0–20 cm soil layer, was the primary contributor to the increased yield of late rice. Moreover, the accumulated temperature and radiation use efficiency, $PEP_N$, WUE, and the net income for MR were higher than those for RR. We conclude that yield increase for the MR cropping system compared with that of the RR cropping system was mainly attributed to the accumulation of biomass post-silking for maize and the accumulation of biomass pre-anthesis for late rice.

**Author Contributions:** Funding acquisition, B.Z.; conceiving and designing the experiment, B.Z. and C.C. (Chuanyong Chen); data collection and writing: original draft, Y.H., D.G. and F.X.; writing: review and editing, W.M., A.S., M.Z. (Ming Zhan), C.C. (Cougui Cao) and M.Z. (Ming Zhao). All authors have read and agreed to the published version of the manuscript.

**Funding:** This work was supported by the Key National Research and Development Program of China (2016YFD0300207) and CAAS Science and Technology Innovation Program (2060302-2).

**Conflicts of Interest:** The authors declare no conflict of interest.

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
