# Peer review of "The Accumulation of Biomass Pre- and Post-Silking Associated with Gains in Yield for Both Seasons under Maize–Rice Double Cropping System"

_agronomy, doi:10.3390/agronomy12061296_

Round 1
Reviewer 1 Report
Dear Authors,
Your manuscript ‘The accumulation of biomass pre- and post-silking associated with gains in yield for both seasons under maize-rice double cropping system’ is very interesting. I believe that the manuscript is of potential interest for readers.
My specific comments, that I hope will help the authors to improve the manuscript:
Line 223 – How was 1000 grains counted? Were 1000 grains counted or were 100 counted and estimated for 1000 grains?
Line 235, 242 … – Check alignment of equation numbering
Line 250 - Visible deformation
Line 262 – Is there any reason why tables and figures do not appear in the text after being cited? I noticed that above, the same thing happens.
Line 502 – I noticed that many of the references are older, if possible replace with newer ones
Author Response
Dear Reviewer,
Thank you for the valuable comments and suggestions concerning our manuscript agronomy-1738284 entitled “The accumulation of biomass pre- and post-silking associated with gains in yield for both seasons under maize-rice double cropping system”. The comments are very helpful to improve the manuscript for publication. We have made a great effort to revise the manuscript in response to your comments and suggestions. The details of modification and revisions are listed as belows.
Sincerely,
Baoyuan Zhou
Resopnse to Comments:
1. Line 223 – How was 1000 grains counted? Were 1000 grains counted or were 100 counted and estimated for 1000 grains?
Response: Thanks for your suggestion. In this research, 1000 grain weight was calculated from the average of five random samples of 500 grains.
2. Line 235, 242 … – Check alignment of equation numbering
Response: Thanks for your suggestion. We have checked the alignment of equation numbering.
3. Line 250 - Visible deformation
Response: Thanks for your suggestion. We have revised the format in the new manuscript.
4. Line 262 – Is there any reason why tables and figures do not appear in the text after being cited? I noticed that above, the same thing happens.
Response: Thanks for your suggestion. We have set all the tables and figures in the text after being cited in the new manuscript.
5. Line 502 – I noticed that many of the references are older, if possible replace with newer ones
Response: Thanks for your suggestion. We have replaced several older references with newer ones in the new manuscript.
Reviewer 2 Report
Dear Authors,
Increasing yields of both corn and rice is of great importance in food production. The use of proper crop rotation improves soil properties in the next growing season, promotes the renewal of soil organic matter and increases nutrient content, increases the activity of soil microorganisms, and promotes the circulation and transformation of nutrients such as nitrogen, phosphorus and potassium in the soil. Therefore, the experiment conducted is of great utilitarian importance. The manuscript is prepared carefully and clearly. The amount of work done on the experiment conducted is immense. The methodology is correct. The work would be richer if the authors showed the changes in the chemical composition of the successor plants.
I have the following comments and suggestions for the presented manuscript:
208 - how long were the samples dried?
2016-2018 - was the energy value of 1 g DM taken or was the sample burned? Was this value obtained e.g. on a calorimetric bomb or taken from tables? If so, on what instrument was this done?
225 - what was the average moisture content of the rice and corn crop?
260 - 274 - since agricultural weather has a huge impact on crop vegetation, therefore weather conditions should be presented for each growing year separately rather than averaged values
275 - 297 - yield results are very important, so it is worth giving them in a table, because exact values cannot be read from drawings and cannot be verified either. The reader is forced to accept the author's interpretation.
296-297 - which table contains the information on this topic?
275- 392 - 3.5 Resource s use efficiency - please provide an example of how the percentages in this chapter were calculated because I am not sure if they are correct both in this chapter and the following chapters.
Author Response
Dear Reviewer,
Thank you for the valuable comments and suggestions concerning our manuscript agronomy-1738284 entitled “The accumulation of biomass pre- and post-silking associated with gains in yield for both seasons under maize-rice double cropping system”. The comments are very helpful to improve the manuscript for publication. We have made a great effort to revise the manuscript in response to your comments and suggestions. The details of modification and revisions are listed as belows.
Sincerely,
Baoyuan Zhou
Resopnse to Comments:
208 - how long were the samples dried?
Response: Thanks for your suggestion. The samples dried to a constant weight, commonly need 14 to 18 hours for maize and 12 to 16 hours for rice.
2016-2018 - was the energy value of 1 g DM taken or was the sample burned? Was this value obtained e.g. on a calorimetric bomb or taken from tables? If so, on what instrument was this done?
Response: Thanks for your suggestion. The energy value was calculated using the gross caloric value according to the method of Li, 2011.
225 - what was the average moisture content of the rice and corn crop?
Response: Thanks for your suggestion. The average moisture content of the rice and corn crop was 14%, and we have clarified it in the manuscript.
260 - 274 - since agricultural weather has a huge impact on crop vegetation, therefore weather conditions should be presented for each growing year separately rather than averaged values
Response: Thanks for your suggestion. We have revised the description of weather conditions, and they were presented for each growing year separately.
275 - 297 - yield results are very important, so it is worth giving them in a table, because exact values cannot be read from drawings and cannot be verified either. The reader is forced to accept the author's interpretation.
Response: Thanks for your suggestion. We have added the yields in Table 3.
296-297 - which table contains the information on this topic?
Response: Thanks for your suggestion. Table 3 contains the information on this topic. However, the sentence description was not very clearly, and we have revised it as “The late rice in both MR and RR cropping systems in 2018 showed significantly higher panicles compared with those in 2017”.
275- 392 - 3.5 Resource s use efficiency - please provide an example of how the percentages in this chapter were calculated because I am not sure if they are correct both in this chapter and the following chapters.
Response: Thanks for your suggestion. For example the production efficiency of Ra for the first season at Wuxue, the averaged value of production efficiency of Ra for the first season of MR and RR over three years was calculated, which was 0.5567 and 0.4433 g MJ-1, respectively, thus the increased of production efficiency of Ra for the first season of MR compared with that under RR was calculated as following: (0.5567-0.4433)/0.4433*100%=25.6%.
Reviewer 3 Report
INCLUDE ALL SUGGESTIONS AND CORRECTIONS THAT ARE MARKED IN THE TEXT OF THE ATTACHED DOCUMENT. J. N. Tabosa - IPA
Author Response
Dear Reviewer,
Thank you for the valuable comments and suggestions concerning our manuscript agronomy-1738284 entitled “The accumulation of biomass pre- and post-silking associated with gains in yield for both seasons under maize-rice double cropping system”. The comments are very helpful to improve the manuscript for publication. We have made a great effort to revise the manuscript in response to these comments and suggestions. The details of modification and revisions are listed as belows.
Sincerely,
Baoyuan Zhou
Resopnse to Reviewer:
- Abstract: INCLUDE IN THE TEXT CULTIVAR NAME. INCLUDE IN THIS ABATRACT WHWRW AND WHEN THE RESEARCH WAS CARRIED OUT.
Response: Thanks for your suggestion. We have added the cultivars name in the section of “2.2. Experimental design and cropping management”.
- Introduction (line 63-66): THIS DISCUSSION USING THESE PERCENTAGE VALUESDOES NOT FIT IN THIS INTRODUCTION. SUGGESTION: TRANSFER TO RESULT AND DISCUSSION.
Response: Thanks for your suggestion. We have transfer the sentence to discussion.
- 1. Introduction (line 75-102): VERY LONG PARAGRAPH. SUGGESTION: REWRIITE USING INDIVIDUALIIZED REFERENCES FOR EACH THEME AND AUTHOR. ALSO HIFHLIGHT THE OBJECTIVE.
Response: Thanks for your suggestion. We have rewritten the paragraph and highlighted the objective in the new manuscript.
- 2. Materials and Methods: DESCIBE THE TREATMENTS AND INFORM THE GENETIC MATERIALS OF MAIZE AND RICE USED.
Response: Thanks for your suggestion. We have added the cultivars name in the section of “2.2. Experimental design and cropping management”.
- 2. Materials and Methods (line 129-130): AVOID LONG PARAGRAPHS. USE SHORT AND OBJETIVE SENTENCESES.
Response: Thanks for your suggestion. We have revised the long sentence to short in the new manuscript.
- 2. Materials and Methods (line 210): ATTENTION TO TEXT FORMATTING AND STANDARDIZING SCIENTIFIC NOTATION.
Response: Thanks for your suggestion. We have revised the text format in the new manuscript.
- 2. Materials and Methods (line 250): CORRECT TEXT FORMATTING
Response: Thanks for your suggestion. We have revised the text format in the new manuscript.
- 3. Results: USE STATISTICAL PARAMETERS IN THE PRESENTATION OF THESE RESULTS, IF APPROPRIATE.
Response: Thanks for your suggestion. We have revised the results according to your advices.
- 4. Discussion (line 402-403): SUGGESTION: TRANSFER THIS INFORMATION TO RESULTS ITEM.
Response: Thanks for your suggestion. We have transferred the information to results.
- 4. Discussion (line 473-475): TRANSFER THIS MARKED PART TO RESULT ITEM.
Response: Thanks for your suggestion. We have transferred the information to results.
- 5. Conclusions (line 483-484): EXTRACT FROM THIS CONCLUSION: SHORT AND OBJECTIVE DIRECT SENTENSES. WITHOUT EXPLAINING: IF IT DOES NOT TURN INTO A DISCUSSION.
Response: Thanks for your suggestion. We have revised the conclusions according to your advices.
- 5. Conclusions (line 489): PUT THE TEXT CONCISELY. ONLY THE RESULTS OF THE DISCUSSION AND WITHOUT EXPLAINING.
Response: Thanks for your suggestion. We have revised the conclusions according to your advices.